# Juvenile Leaf Color Changes and Physiological Characteristics of *Acer tutcheri* (Aceraceae) during the Spring Season

Yating Xie [1,2], Nancai Pei [1], Zezhou Hao [1], Zhaowan Shi [1], Lei Chen [3,*], Baoying Mai [4], Qunhui Liu [5], Jiajie Luo [5], Mingdao Luo [4] and Bing Sun [1,*]

1 Research Institute of Tropical Forestry, Chinese Academy of Forestry, Guangzhou 510520, China
2 College of Landscape Architecture, Nanjing Forestry University, Nanjing 210037, China
3 International Centre for Bamboo and Rattan, Beijing 100102, China
4 Research Institute of Forestry in Gaoming District, Foshan 528000, China
5 Urban Forestry Station in Gaoming District, Foshan 528000, China
* Correspondence: chenlei@icbr.ac.cn (L.C.); gdsunbing@126.com (B.S.); Tel.: +86-10-84789935 (L.C.); +86-20-87033625 (B.S.)

**Abstract:** Leaf color is a key trait that determines the ornamental quality of landscape tree species such as *Acer tutcheri*, and anthocyanin is the main pigment for red leaf coloration. Red leaf fading significantly reduces the ornamental value of *A. tutcheri* leaves in the spring; however, the physiological mechanism that causes red leaf discoloration in this species is still unclear. Only the anabolic or degradative metabolism of anthocyanin has been studied in terms of leaf color changes. In this study, leaves from four color-change stages of *A. tutcheri* during the spring were selected by the average sampling method, which involves measuring and analyzing the pigment content and physiological factors related to anthocyanin metabolism, aiming to clarify the key physiological factors of spring leaf coloration. Our results show that the reduced anthocyanin/chlorophyll ratio was the direct cause of red leaf fading in spring. The carbohydrates synthesized during the growth of juvenile leaves were mainly used for their growth and development, whereas fewer carbon sources were used for the synthesis of anthocyanin. Phenylalanine ammonia-lyase and chalcone isomerase activities increased in the early stages of juvenile leaf development and decreased in the middle and late stages, whereas peroxidase activity continued to increase. The decrease in anthocyanin synthesis-related enzyme activity reduced the accumulation of anthocyanin, whereas the increase in anthocyanin degradation-related enzyme activity accelerated the depletion of anthocyanin. Increasing vacuole pH was a major factor in the degradation of anthocyanin. The physiological characteristics of *A. tutcheri* leaves during the spring season suggest that anthocyanin is a key factor affecting the presentation of color in juvenile leaves, and red leaf fading in the spring of *A. tutcheri* is the result of the joint effect of the decrease in anthocyanin anabolic metabolism and the increase in anthocyanin degradative metabolism.

**Keywords:** *Acer tutcheri*; juvenile leaves; anthocyanin; physiological characteristics; color fading





## 1. Introduction

*Acer tutcheri* is a typical landscape tree of the maple genus *Acer* (Aceraceae). The juvenile leaves sprouting in the spring are bright scarlet, summer maturity leaves are bright green, and the leaves change to bright red or orange–red in the autumn and deep red in the winter. *Acer tutcheri* is an excellent native landscape tree species with obvious seasonal ornamental characteristics [1]. Currently, research on the leaf color of landscape species such as *A. tutcheri* focuses primarily on autumn leaves [2], and studies on spring leaf color are mostly concerned with interpreting the ecological function significance of juvenile leaf color [3,4], whereas there is a dearth of research on the physiological mechanism of *A. tutcheri*'s spring leaf color change.

The most significant change observed during the spring leaf growth of the plant is the change in leaf color, with the juvenile leaves of the new tips of many plants initially

appearing as red, pink, and other colors. The leaf color changes as the leaf age increases [5,6]. Changes in the type, content, and distribution of pigment directly cause changes in leaf color [7]. Anthocyanin is a key factor affecting the presentation of color in juvenile leaves [8]. Anthocyanin is an important class of flavonoid pigments in plant leaves that is formed naturally from anthocyanidins and various glycosyl groups; it often accumulates in the epidermal or subepidermal cells of leaves, as well as in palisade and spongy tissues [9,10]. The absolute content of anthocyanin is a critical factor in the presentation of leaf color, and the level of anthocyanin content is determined by a combination of anthocyanin synthesis, degradation, and stability [9].

As part of the phenylpropanoid pathway, the biosynthesis of the anthocyanin has been well characterized. Under the regulation of transcription factors such as MYB, bHLH, and WD40, phenylalanine in the cytoplasm is synthesized into colored anthocyanidin molecules by the action of enzymes including phenylalanine ammonia-lyase (PAL), chalcone synthase (CHS), chalcone isomerase (CHI), and anthocyanidin synthesis (ANS) [11,12]. Due to the highly unstable structure of free anthocyanidin, it combines with glycosyl groups to form glycosides known as anthocyanin [12]. The anthocyanin in the cytoplasm is then transported to the vacuole for storage by transport proteins such as glutathione S-transferase (GST) [13]. When a link in the anthocyanin biosynthetic pathway is disturbed or inhibited, it affects the overall efficiency and feedback regulation of the pathway, resulting in changes in the color of anthocyanin. The specific PAL inhibitor aminooxy phenyl propionic acid (AOPP) was reported to reduce the content of anthocyanin in detached flower buds, and it was hypothesized that decreasing PAL activity would inhibit anthocyanin synthesis in specific developmental stages of *Petunia hybrida* [14]. According to Nakamura et al., suppression of petunia ANS activity causes an increase in the number of white blooms [15]. In addition to serving as a precursor for anthocyanin synthesis, sugar serves as a signaling regulatory molecule for enzymes involved in anthocyanin synthesis, including PAL, CHS, and ANS [16,17]. During the last stages of *Paeonia suffruticosa* flower development, glucose regulates anthocyanin biosynthesis through signaling to induce the expression of genes such as PSBHLH3 [18].

Compared to anthocyanin synthesis, there are fewer studies on the stability and degradation metabolism of anthocyanin. The level of enzyme activity and the properties of the vacuole environment are thought to be related to the degradation and stability of anthocyanin. Peroxidase (POD) is reported to be an enzyme that uses anthocyanin as a substrate, and anthocyanin loses its color through peroxidase oxidation [14]. According to Zipor et al., the peroxidase BCPRX01 was co-localized with anthocyanin in the vacuoles of *Brunfelsia latifolia* petals, and the mRNA and protein levels were significantly induced during the degradation of anthocyanin [19]. High temperatures and intense light are examples of environmental stressors that can accelerate the degradation of anthocyanin by activating peroxidase activity [20,21]. The potential role of polyphenol oxidase (PPO) in anthocyanin degradation has also been demonstrated. Polyphenol oxidase activity increases during the browning of *Litchi chinensis* fruit, and when its activity is inhibited in the litchi pericarp, anthocyanin degradation is delayed [22,23]. On the other hand, the stability of anthocyanin is influenced by vacuole pH conditions and natural co-pigmentation substances [24]. When pH < 3, anthocyanin is more stable, whereas an increase in pH decreases its stability and causes anthocyanin degradation [25,26]. Luo et al. found that tannins, a co-pigmentation substance in *Rosa chinensis* petals and *Excoecaria cochinchinensis* leaves, significantly inhibit anthocyanin-degrading enzymes, preventing anthocyanin from being degraded [27,28]. These findings show that carbohydrates, anthocyanin synthesis-related enzymes, polyphenol oxidases, peroxidases, vacuole pH, and tannin compounds all play parts in controlling plant color. They are strongly tied to the physiological metabolic levels of anthocyanin in various plants and have significant roles in plant color changes.

As test subjects for this study, we selected the leaves of *A. tutcheri*, which presented various colors at different leaf ages in spring. By measuring the content of pigment, osmotic material, and physiological factors related to anthocyanin metabolism in leaves

at each stage, in this study, we investigated the color performance features and related physiological properties in the spring leaf growth phase of *A. tutcheri* to determine the major physiological factors on leaf color presentation. This study sheds light on the formation of differently colored leaves in *A. tutcheri* and provides a reference for reasonable cultivation measures to improve its decorative characteristics.

## 2. Materials and Methods

### 2.1. Experimental Design

The test material was obtained from 8-year-old *Acer tutcheri* (Mount Qiniang lineage No. 1), which were planted in the Planting Demonstration Park in Mount Ludong (22°80′ N, 112°67′ E; average altitude of 130 m), Foshan City, Guangdong Province, China. Referring to Yin's method [29], the diameter at breast height (DBH) and the tree height of the *A. tutcheri* in the planting area were measured, the mean value was calculated, and nine *A. tutcheri* trees with DBH and tree height close to the mean value (Table 1) were selected as the research objects. Leaf color was observed beginning in late March 2022, and samples were taken from each direction of east, south, west, and north of each selected tree. Sampling was conducted at intervals of 15 days for four consecutive sampling sessions according to the developmental status of leaves and the rate of leaf color change. In each period, eight leaves were collected from each of the chosen branches. The leaves were characterized as having an all-red leaf (24 March 2022), a reddish area of about ninety percent (8 April 2022), a reddish area of about fifty percent (23 April 2022), and an all-green leaf (8 May 2022), respectively (Figure 1). To ensure the normal and healthy growth of *A. tutcheri* and to meet the number of leaves required for the experiment, 8–10 leaves were collected randomly from selected branches in each period, and the leaves of every three trees were mixed as one biological replicate for a total of three biological replicates. One part of the collected leaves was wrapped in tin foil and stored in liquid nitrogen; they were analyzed for physiological indicators. The other part was immediately cut along the main vein into 5 mm long and 2 mm wide pieces and immersed in FAA fixative solution (45% anhydrous ethanol, 6% acetic acid, and 5% formaldehyde) purchased from Beijing Coollaber Science & Technology Co. Ltd., Beijing, China (catalog number: SL1620-500 mL).

**Table 1.** Information about the nine *Acer tutcheri* trees.

| Number | DHB (cm) | Height (m) | Crown Diameter (m) | Under-Branch Height (m) |
|--------|----------|------------|--------------------|--------------------------|
| QN.1 | 3.98 | 2.95 | 2.20 | 1.35 |
| QN.2 | 3.92 | 3.04 | 2.32 | 1.33 |
| QN.3 | 4.00 | 3.10 | 2.40 | 1.50 |
| QN.4 | 4.12 | 2.86 | 2.56 | 0.92 |
| QN.5 | 4.10 | 3.21 | 2.17 | 1.63 |
| QN.6 | 4.05 | 2.96 | 2.65 | 1.13 |
| QN.7 | 4.20 | 2.82 | 2.50 | 1.00 |
| QN.8 | 4.15 | 3.22 | 2.42 | 1.20 |
| QN.9 | 3.98 | 3.12 | 2.55 | 1.35 |

QN refers to the "Mount Qiniang provenance, lineage No. 1".

### 2.2. Determination of Physiological Indicators

Chlorophyll (Chl) and carotenoid (Car) were determined according to Lichtenthaler and Wellburn [30,31] as a reference. The leaves were cleaned with distilled water, dried with absorbent paper, cut into 1 mm wide and 2 mm long thin filaments using scissors, and mixed with the thin filaments from different leaves from the same sample. Then, 0.1 g of the leaves was weighed, and 10 mL of a solution made up of 95% ethanol and 80% acetone was added, sealed, and placed in a constant temperature incubator at 32 °C. They were then extracted away from the light for about 24 h until they turned white or transparent. Using the 95% ethanol and 80% acetone mixture as a blank control, the absorbance (A) of the extracts at 470 nm, 646 nm, and 663 nm was measured under a UV-T6 spectrophotometer (Beijing Purkinje General Instrument Co. Ltd., Beijing, China).

Using the formula, we calculated the content and concentration of carotenoid (Car), total chlorophyll (Chl), chlorophyll a (Chl a), and chlorophyll b (Chl b). Chlorophyll a ($C_a$) content was calculated as $(12.21 \times A_{663} - 2.81 \times A_{646})/W$ mg g$^{-1}$ of fresh weight (FW). Chlorophyll b ($C_b$) content was calculated as $(20.13 \times A_{646} - 5.03 \times A_{663})/W$ mg g$^{-1}$ FW. Total chlorophyll content ($C_T$) was calculated as ($C_a + C_b$). Carotenoid content ($C_c$) was calculated as $(1000 \times A_{470} - 3.27 \times C_a - 104 \times C_b)/W$ mg g$^{-1}$ FW, where "W" is an abbreviation for "weight" and "FW" is an abbreviation for "fresh weight".

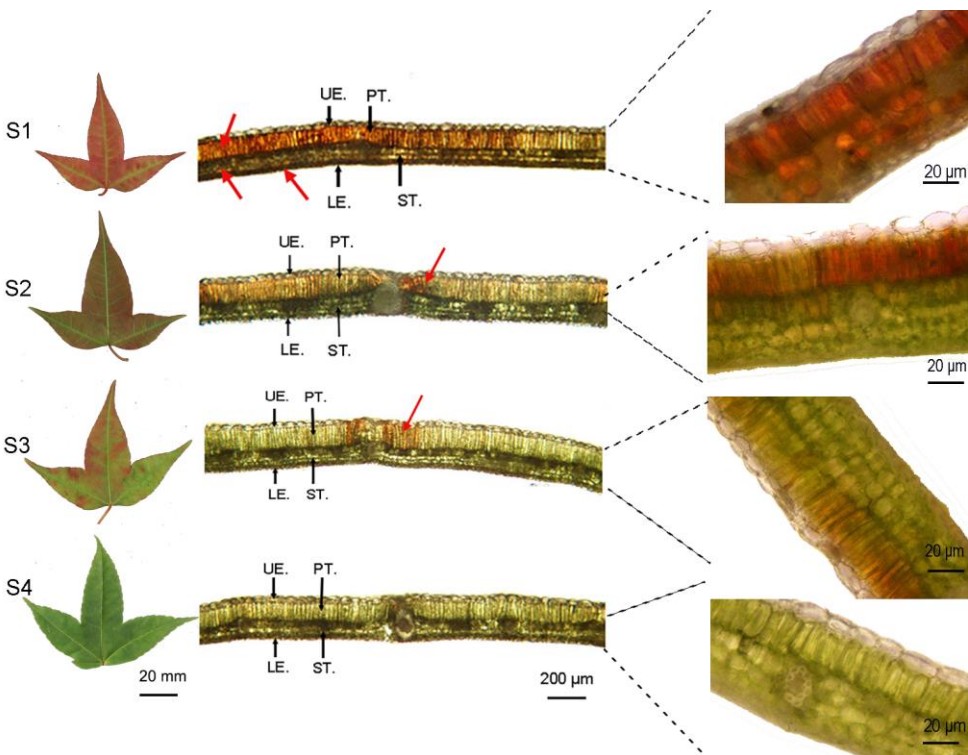

**Figure 1.** Color characteristics and pigment distribution of leaves of *Acer tutcheri* at different coloration periods of spring. S1: 24 March 2022; S2: 8 April 2022; S3: 23 April 2022; S4: 8 May 2022. UE: upper epidermis; LE: lower epidermis; PT: palisade tissue; ST: spongy tissue. The red arrow points to the anthocyanin distribution position.

To determine anthocyanin (Ant) content [32], the leaves were washed with distilled water, dried with absorbent paper, cut into 1 mm wide and 2 mm long thin filaments using scissors, and mixed with the thin filaments from different leaves from the same sample. Then, 0.1 g of leaves was weighed, and 10 mL of a solution made up of 1% hydrochloric acid and anhydrous ethanol was added, sealed, and placed in a constant temperature incubator at 32°C for 8 h of extraction away from light. Using the 1% hydrochloric acid and anhydrous ethanol mixture as a blank control, the absorbance (A) of the extracts at 530 nm and 657 nm was measured under a UV-T6 spectrophotometer (Beijing Purkinje General Instrument). We then calculated the concentration and content of anthocyanin (Ant) [28]. Anthocyanins content was calculated as $(A_{530} - 0.25A_{657})/W$ mg g$^{-1}$ (FW).

The pH was determined as described by Oren-Shamir et al. [33] by taking 1 g of leaf sample and grinding it at a ratio of 0.1 g tissue to 1 mL $H_2O$ to obtain homogenate and measuring the change in cell sap pH with a PHSJ-3F acidometer (Shanghai INESA Scientific Instrument Co. Ltd., Shanghai, China).

Soluble sugar content (catalog number: KT-1-Y), soluble protein content (catalog number: KMSP-1-W), tannin content (catalog number: DN-1-Y), phenylalanine ammonia-lyase activities (catalog number: PAL-1-Y), chalcone isomerase activities (catalog number: CHI-1-Y), peroxidase activities (catalog number: POD-1-Y), and polyphenol oxidase activities (catalog number: POD-1-Y) were measured according to the biochemical test kits provided

by Suzhou Comin Biotechnology Co. Ltd., China. Each physiological index was repeated three times.

### 2.3. Preparation of Freehand Sectioning, Paraffin Sectioning, and Staining by Periodic Acid–Schiff (PAS)

#### 2.3.1. Freehand Sectioning Method

The middle cross section of the leaf was sliced with two tightly pressed blades placed side by side, quickly immersed in distilled water, dipped with a brush, placed on a slide with a drop of distilled water, and covered with a coverslip. The distribution of chlorophyll and anthocyanin in leaf tissue was observed and photographed under a light microscope (Primo Star, Zeiss, Germany).

#### 2.3.2. Staining Carbohydrates in the Leaf by Periodic Acid–Schiff (PAS) Staining

The fixed samples were prepared in paraffin sections with a section thickness of 3 um according to the method described by Wang [34], and three sections were made for each leaf. Dewaxing was conducted as follows: Xylene I for 20 min, xylene II for 20 min, 100% ethanol I for 5 min, 100% ethanol II for 5 min, and 75% ethanol for 5 min, followed by rinsing with tap water. The sections were stained using a PAS dye solution set (G1008, Servivebio). First, sections were stained with 0.5% periodate acid solution for 10–15 min, and then rinsed with tap water and twice with distilled water. Then, sections were stained with Schiff reagent for 25–30 min in the dark and rinsed for 5 min. Dehydration was conducted as follows: 100% ethanol I for 5 min, 100% ethanol II for 5 min, 100% ethanol III for 5 min, xylene I for 5 min, and xylene II for 5 min. Finally, the samples were sealed with neutral balsam. The sections were observed and photographed using a Zeiss optical microphotography system (ZEISS Imager. M2, Germany) with 5 fields of view per section.

### 2.4. Date Statistical and Analysis

Data collation, one-way ANOVA, Duncan's multiple range test, and calculation of the path coefficient were performed using Excel 2022, SPSS 26.0, and MATLAB R2021b. Plots were generated using OriginPro 2022 and SPSS_Amos_24. Bivariate correlation analysis was performed using the Pearson correlation test. Three biological replicate samples were used to gather data. Values are presented as mean $\pm$ standard error of the mean (SEM).

Path analysis: Following the methods described by Douglas R. Dewey [35] and Yuan [36,37] according to the principle of path analysis, we used the correlation coefficient to create a system of equations as follows:

$$\begin{cases} p_{x1y} + r_{x1x2} \times p_{x2y} + \ldots + r_{x1xj} \times p_{xiy} = r_{x1y} \\ r_{x2x1} \times p_{1y} + p_{x2y} + \ldots + r_{x2xj} \times p_{xiy} = r_{x2y} \\ \vdots \\ r_{xjx1} \times p_{1y} + p_{xjy} \ldots + r_{xixj} \times p_{xiy} = r_{xjy} \end{cases}$$

where $r_{xjx1}$ denotes the simple correlation coefficient between xi and xj, $r_{xiy}$ denotes the correlation coefficient (total effect) between the independent variable (xi) and the dependent variable (y), $p_{xiy}$ denotes the direct path coefficient (direct effect) between xi and y, and $p_{yxj}$ denotes the indirect path coefficient (indirect effect) of xi through xj on y. $p_{yxj} = r_{xixj} \times p_{xiy}$, $r_{xiy} = p_{xiy} + \sum p_{yxj}$. The direct path coefficient ($p_{xiy}$) can be obtained using MATLAB R2021b. The absolute values of the direct and indirect correlation coefficients indicate the magnitude of the direct and indirect effects, respectively. The correlation coefficient between each independent variable and the dependent variable is the total effect.

$R_i^2 = 2 \times p_{xiy} \times r_{xiy} - p_{xiy}^2$, $R_i^2$ is the decision-making coefficient, which indicates the combined deterministic effect of the independent variable (xi) on the dependent variable (y) through x1, x2, ..., and xj. The key elements can be determined by ranking the significance of the combined effects.

## 3. Results

### 3.1. Characteristics of the Leaf Pigment Content, Ratio, and Distribution

The results of slicing show that both the upper and lower epidermis of the leaf consist of interdigitated monolayers, with no anthocyanin distribution in the upper epidermal cells in S1 to S4 periods, whereas the lower epidermis had anthocyanin distribution only in the S1 period (Figure 1). The palisade tissue was the main site of anthocyanin and chlorophyll distribution, and the red color kept lightening and the green color kept deepening during periods S1 to S4. The spongy tissue has wide cellular gaps with numerous air cavities and only a small amount of anthocyanin distribution in the S1 period, and the rest of the periods were green.

The content of pigments in the leaves of *Acer tutcheri* differed significantly during different leaf coloration periods (Figure 2). As the leaves turned from red to green, the levels of Chl a, Chl b, and Car increased, but Ant levels declined. In the S1 period, the contents of Chl a, Chl b, and Car were 0.347 mg/g, 0.089 mg/g, and 0.183 mg/g, respectively, and in the S4 period, the contents were 1.317 mg/g, 0.378 mg/g, and 0.430 mg/g, respectively. In comparison to the S1 period, the contents of Chl a, Chl b, and Car increased by 279.5%, 324.7%, and 135.0%, respectively. Ant content decreased significantly, from 9.372 mg/g in the S1 period to 0.805 mg/g in the S4 period, a loss of more than 91.4%.

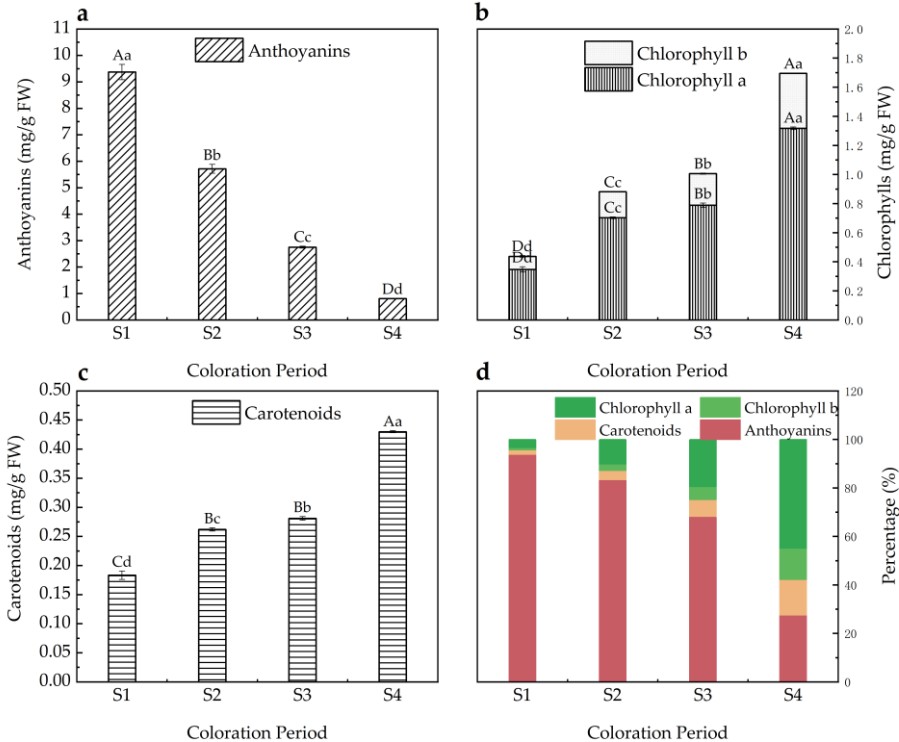

**Figure 2.** The content and proportion of photosynthetic pigments and anthocyanin in juvenile *Acer tutcheri* leaves during different coloration periods in spring. (**a**) Anthocyanin (Ant) content. (**b**) Chlorophyll a (Chl a) and Chlorophyll b (Chl b) content. (**c**) Carotenoid (Car) content. (**d**) Pigment proportion. Different capital letters indicate a significant difference at $p < 0.01$, and different lowercase letters indicate a significant difference at $p < 0.05$.

As the red color of the leaves faded, the proportion of each pigment also changed. During the S1 period, the proportion of Ant found in the leaves was 93.8%, whereas the proportion of (Chl + Car) was only 6.2%. The Ant content was 15.13 times higher than the (Chl + Car) content. By the S4 period, the percentage of Ant content had gradually reduced to 27.5%, whereas the percentage of (Chl + Car) content had gradually increased to 72.5% (Figure 2d). The results of the changes in anthocyanin content and chlorophyll content are consistent with the distribution of the pigments in the slices (Figure 1).

### 3.2. Changes in PAL Activity, CHI Activity, Vacuole pH and Tannin Content, Soluble Sugar Content, Starch Content, and Soluble Protein Content

The trend of PAL and CHI activities was consistent during different leaf coloration periods, with a single-peaked curve that first increased and then decreased, increasing from between the S1 to S2 periods, with the highest activity in the S2 period, with 710.912 U/g and 673.409 U/g, respectively. Then, it continued to decrease and fell to 477.640 U/g and 496.679 U/g during the S4 period, a decrease of 19.8% and 25.0%, respectively, compared to the S1 period (Figure 3).

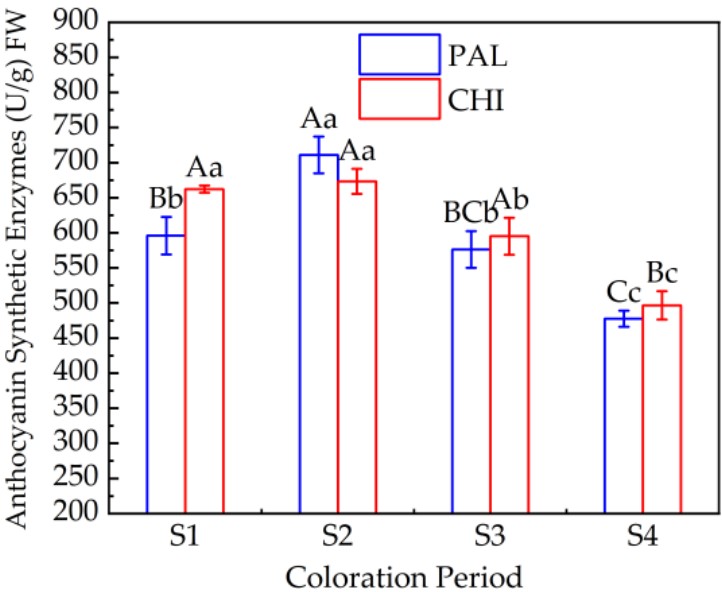

**Figure 3.** Enzyme activity related to anthocyanin synthesis in juvenile *Acer tutcheri* leaves during different coloration periods in spring. Different capital letters indicate a significant difference at $p < 0.01$, and different lowercase letters indicate a significant difference at $p < 0.05$.

As shown in Figure 4, the contents of soluble sugars and starch increased gradually as the leaves reddened, and the contents of both were significantly higher in the S2, S3, and S4 periods than in the S1 period ($p < 0.01$) and increased slightly in the S2–S4 periods; however, the differences were not significant. The soluble sugar content was higher than the starch content in in periods S1 to S4. The soluble protein content remained essentially at a similar level in each period.

To further clarify the changes in carbohydrate content during leaf development, the content and distribution of carbohydrates in leaves were observed by staining with PAS. The results show (Figure 4c) that carbohydrates were mainly distributed in the palisade tissue during periods S1 to S4, which is consistent with the changes in the contents of soluble sugars and starch shown in Figure 4a,b, respectively.

### 3.3. Changes in POD Activity, PPO Activity, Vacuole pH, and Tannin Content

Our results show that POD activity continued to increase during periods S1 to S4, with a sharp increase to 41.497 U/g during the S4 period and a 148.0% increase in POD activity during the S4 period compared to the S1 period, with the difference in the rate of increase between periods reaching a highly significant level. The changes in PPO activity from S1 to S4 were all minor and not significantly different (Figure 5).

As shown in Figure 6a, the vacuole pH gradually increased from period S1 to period S4, with pH values of 3.743, 3.810, 3.867, and 3.927, respectively; the differences reached a highly significant level in each of the periods ($p < 0.01$). The tannin contents in periods S1 to S4 were 2.664 mg/g, 2.730 mg/g, 2.718 mg/g, and 2.739 mg/g, respectively, and did not reach significant differences in any period (Figure 6b).

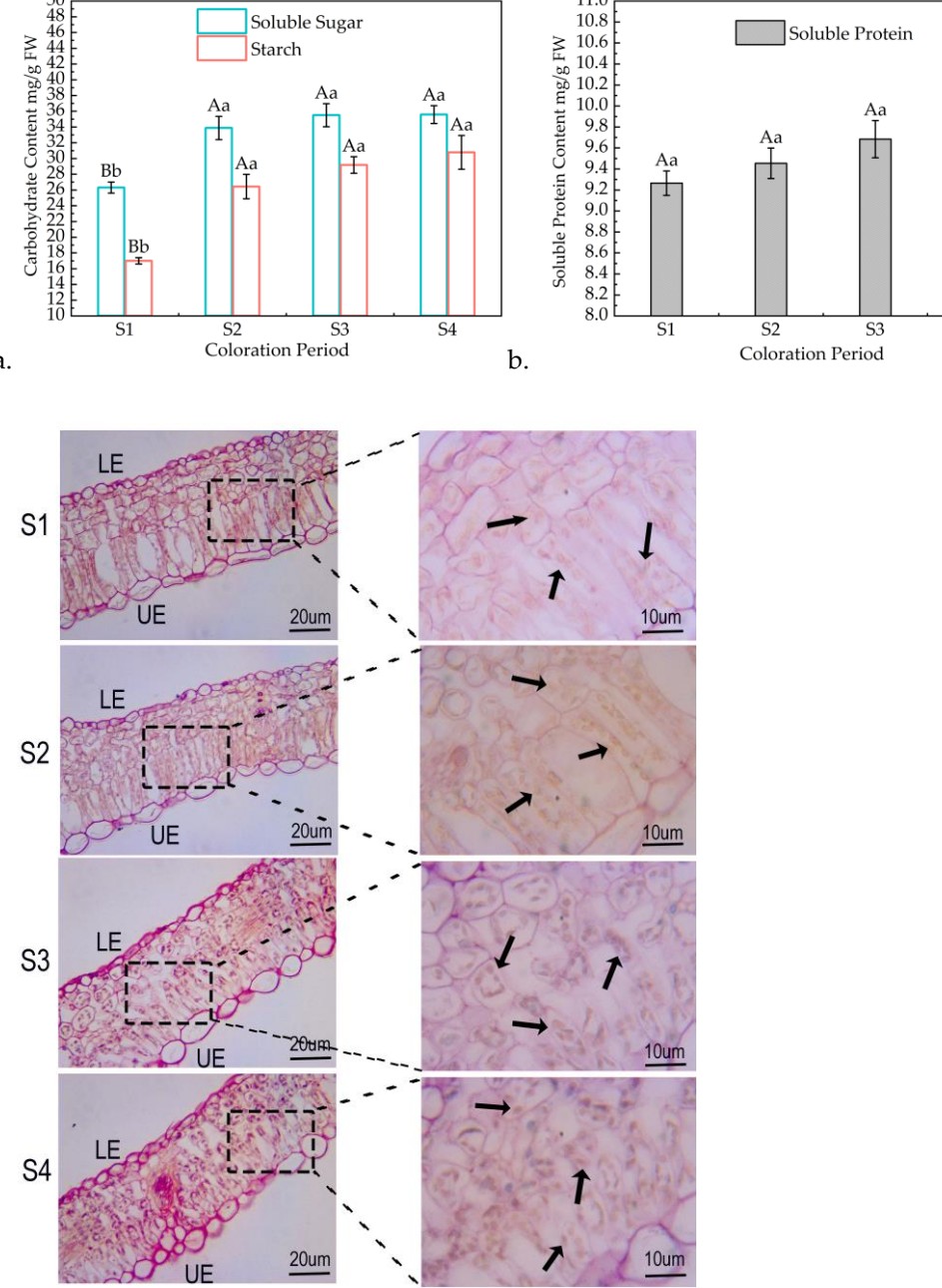

**Figure 4.** Content and distribution of soluble sugar, starch, and soluble protein in juvenile *Acer tutcheri* leaves during different coloration periods in spring. (**a**) Soluble sugar and starch content. (**b**) Soluble protein content. (**c**) Distribution and content of carbohydrates; the small red–violet dot with the black arrow indicates carbohydrates stained by periodic acid–Schiff (PAS). Different capital letters indicate a significant difference at $p < 0.01$, and different lowercase letters indicate a significant difference at $p < 0.05$.

### 3.4. Correlation Analysis of Anthocyanin with Physiological Factors and Path Analysis

Leaf pigment level and physiological factors were correlated to varying degrees (Figure 7a). There was a highly significant positive correlation between anthocyanin content and chalcone isomerase activity. On the contrary, there were significant negative correlations between anthocyanin and chlorophyll content, carotenoid content, soluble sugar content, starch content, peroxidase activity, and vacuole pH. There was no significant correlation between anthocyanin and soluble protein, tannin, and phenylalanine ammonia-lyase.

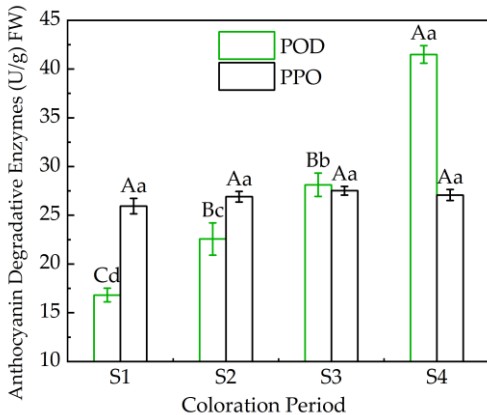

**Figure 5.** Enzyme activity related to anthocyanin degradation in juvenile *Acer tutcheri* leaves during different coloration periods in spring. Different capital letters indicate a significant difference at $p < 0.01$, and different lowercase letters indicate a significant difference at $p < 0.05$.

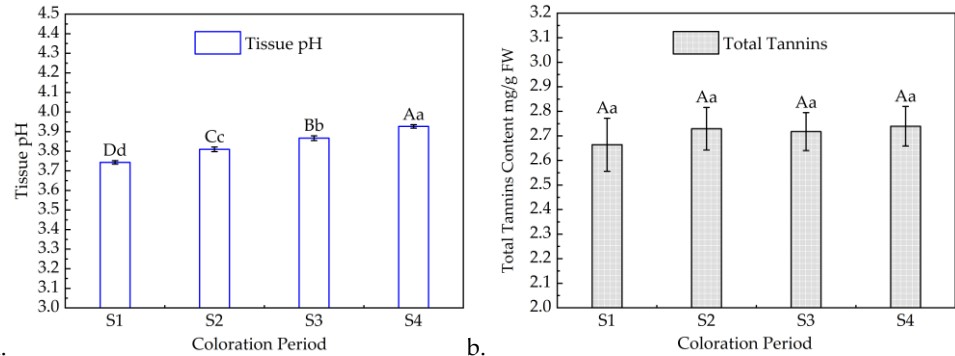

**Figure 6.** Tissue pH and total tannin content in juvenile *Acer tutcheri* leaves during different coloration periods in spring. (**a**) Change in tissue pH. (**b**) Change in total tannin content; different capital letters indicate a significant difference at $p < 0.01$, and different lowercase letters indicate a significant difference at $p < 0.05$.

To better illuminate the magnitude of the direct and indirect effects of each factor on anthocyanin, path analysis was carried out using anthocyanin as the dependent variable and PAL, CHI, POD, PPO, vacuole pH, and Tan as independent factors. As shown in Figure 7b and Table 2, the total effect magnitude of each factor on anthocyanin (in descending order) was pH > POD > CHI > PAL > PPO > TAN; the direct effect magnitude was pH > POD > PAL > CHI > PPO > TAN; and the decision coefficient was pH > POD > PPO > TAN > CHI > PAL. The three rankings were not in the same order of magnitude, suggesting that indirect effects also play an important role in anthocyanin metabolism. The total effect of vacuole pH on the effect of anthocyanin was −0.916, and the direct path coefficient was −0.876, whereas the sum of indirect path coefficients was small. The effect of vacuole pH on anthocyanin content was mainly negative and direct, and the decision-making coefficient of vacuole pH was the largest (0.937), indicating that vacuole pH played a dominant role in anthocyanin content. The total effect of the POD on anthocyanin was −0.916, with a direct path coefficient of only −0.233. The decision-making coefficient was 0.372, ranking second, and POD influenced anthocyanin content but mainly affected the results through the indirect effect of vacuole pH. The direct path coefficients of PAL and CHI were negative (−0.149), but their indirect path coefficients through vacuole pH (0.546, 0.748) and POD (0.154, 0.203) were greater, offsetting the negative direct effects, and the effects of PAL and CHI on anthocyanin were dominated by indirect effects. The total effect and direct effect of PPO and Tan on anthocyanin action were small.

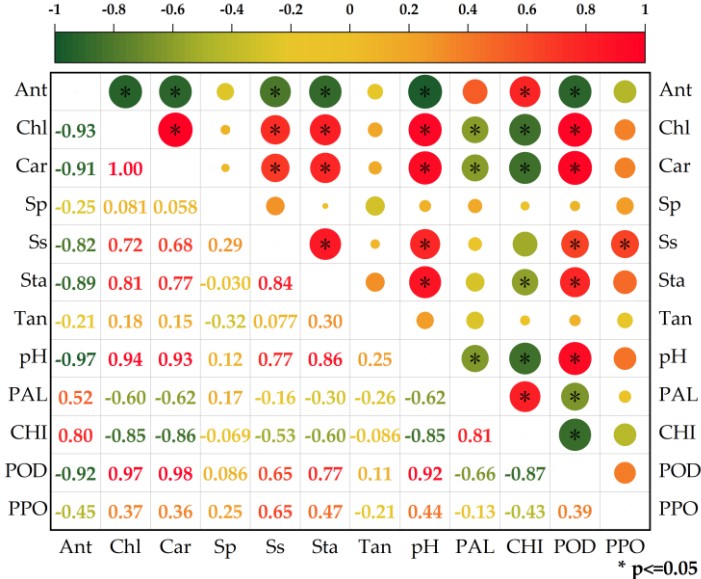

a.

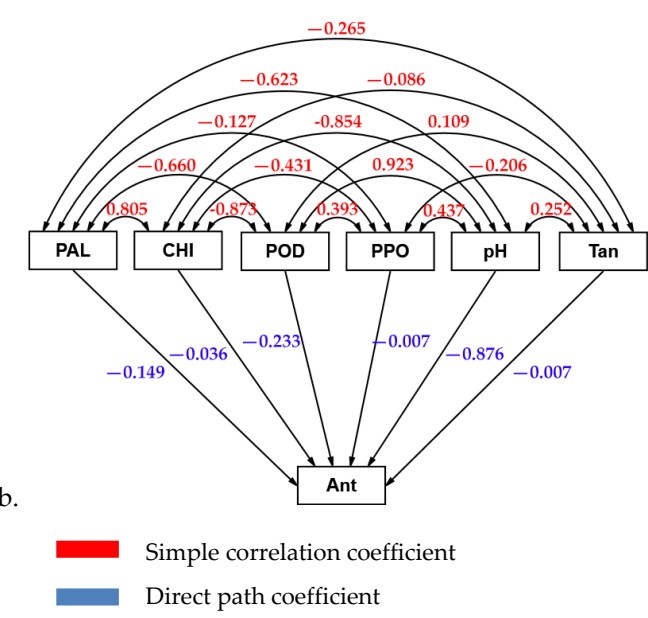

b.

■ Simple correlation coefficient

■ Direct path coefficient

**Figure 7.** Correlation analysis and path analysis between anthocyanin and physiological factors. (**a**) Correlation analysis. (**b**) Path analysis.

**Table 2.** Path coefficient between anthocyanin and physiological factors.

| Factor | Total Effect | Indirect Path Coefficient | | | | | | Decision-Making Coefficient |
|---|---|---|---|---|---|---|---|---|
| | | By PAL | By CHI | By POD | By PPO | By pH | By Tan | |
| PAL | 0.525 | | −0.029 | 0.154 | 0.001 | 0.546 | 0.002 | −0.178 |
| CHI | 0.800 ** | −0.120 | | 0.203 | 0.003 | 0.748 | 0.001 | −0.059 |
| POD | −0.916 ** | 0.098 | 0.031 | | −0.003 | −0.809 | −0.001 | 0.372 |
| PPO | −0.446 | 0.019 | 0.015 | −0.091 | | −0.383 | 0.001 | 0.006 |
| pH | −0.973 ** | 0.093 | 0.031 | −0.215 | −0.003 | | −0.002 | 0.937 |
| Tan | −0.209 | 0.039 | 0.003 | −0.025 | 0.001 | −0.221 | | 0.003 |

** indicates a highly significant correlation between anthocyanin and this index ($p < 0.01$).

## 4. Discussion

*4.1. Pigment Distribution Characteristics, Content, and the Ratio of Leaves during Different Periods of Color Change*

In many landscape trees, juvenile leaves often appear red in the spring, and as the leaves mature, they turn completely green [6,38]. Changes in the type, proportion, and spatial distribution of pigments determine the variation in leaf color [7]. The same is true for the characteristics of the leaf color change of *A. tutcheri* in spring. From period S1 to period S4, the red pigment, which was distributed in several spaces, such as the lower epidermis, palisade tissue, and spongy tissues, gradually decreased as was only partially distributed in the palisade tissue; on the contrary, the distribution space of green pigment gradually increased (Figures 1 and 2). The content and proportion of anthocyanin decreased continuously, and the content and proportion of chlorophyll and carotenoid increased gradually. During this process, although elongation and growth of leaf cells result in a slight dilution of anthocyanin concentration [39], the fading red color is mainly due to the reduction in anthocyanin content, proportion, and distribution space; the same color change occurs in *Prunus* [40], *Acer rubrum*, and *Acer truncatum* [41]. The anthocyanin metabolism in juvenile leaves is closely related to the developmental process of the leaves, and immature leaves are more vulnerable to light stress compared to mature leaves [42]. Anthocyanin protects juvenile leaves with low light processing and carbon fixation capacity by absorbing excess visible and ultraviolet light, allowing photosynthesis to proceed smoothly in the chloroplast system [43]. As the leaves mature, the leaf epidermis gradually forms a photoprotective waxy layer, and the organelles are also developed to maturity; the mature leaves no longer require anthocyanin for photoprotection, anthocyanin degradation is actively increased, and the color of the leaves turns from red to green as they mature [44], as it is also proven by the sequential disappearance of anthocyanin from the lower epidermis, palisade tissue, and spongy tissue.

*4.2. Analysis of Physiological Factors Related to Anthocyanin Synthesis during Different Leaf Color Periods*

A combination of a loss of anabolic metabolism of anthocyanin and an increase in its degradative metabolism causes the red color of leaves to fade in the spring [9,45]. Previous studies indicated that significant downregulation of anthocyanin-synthesizing enzyme genes such as MpCHS and MpDFR and upregulation of anthocyanin-degrading enzyme genes such as MpPOD1 and MpPOD8 in *Malus spectabilis* and *Prunus salicina* resulted in fruit discoloration [45,46]. In this study, PAL activity greatly increased in the S1 to S2 period, but PAL and CHI activity significantly decreased in the S2 to S4 period (Figure 3). This may be because PAL is the first enzyme in the phenylpropanoid pathway and is involved in both the anthocyanin synthesis pathway and the metabolism of other branches of the phenylpropanoid pathway [47]. During the S1 and S2 periods, when the leaves of *A. tutcheri* were not fully developed and various metabolic pathways were active, the PAL-catalyzed reaction produced precursors not just for the synthesis of anthocyanin but also for the synthesis of other compounds, which led to a significant increase in PAL activity in this period. CHI is the second key enzyme in the anthocyanin synthesis pathway [11]. The activities of PAL and CHI were significantly reduced from S2 to S4, presumably due to inhibition of the anthocyanin synthesis pathway metabolism, and the red color of the leaves ultimately faded.

Carbohydrates, as part of the anthocyanin structure, provide precursor substances, energy sources, and osmoregulation for anthocyanin synthesis. The carbohydrate content in the leaves affects the accumulation of anthocyanin. It has been proven that sucrose [48], glucose [18], and fructose [49] are the major carbohydrates that actively promote the accumulation of anthocyanin. Through specific signal transduction pathways, carbohydrates can act as signaling molecules to regulate the expression of genes involved in anthocyanin synthesis and to induce anthocyanin synthesis [18,50]. In this study (Figure 4), as the juvenile leaves of *A. tutcheri* matured, the contents of soluble sugars and starch increased

significantly compared to the S1 period, but the anthocyanin content decreased significantly, and the correlation analysis showed a significant negative correlation with anthocyanin content, in contrast to the results reported in the abovementioned study. The carbohydrate content of juvenile leaves increased, and the anthocyanin content decreased; this may be related to the transition of juvenile leaves from the sink system to the source system during leaf development [51]. Juvenile leaves and other newborn tissues are the most important sink system of plants, and mature leaves are the suppliers of photosynthetic products to plants and the most important source system [52]. The correlation between sugars and anthocyanin can be decoupled by source-sink regulation [53]. According to the carbon-nutrient balance (CNB) hypothesis mentioned by Bobeica et al., phenylpropyl-derived compounds such as anthocyanin are strongly dependent on carbon in leaves, and when leaves have limited carbon sources, carbon sources are preferentially allocated to sugar accumulation to meet their growth and developmental needs, thereby allocating fewer available carbon sources to secondary metabolite pathways [54]. The carbon overflow hypothesis also mentions that when the carbohydrate content in the leaf is higher than the demand of the leaf itself, the "excess" carbohydrate exists in the form of carbon flow through the phenylpropanoid pathway, with anthocyanidin as an additional carbon library, forming anthocyanin; only then is the sugar content positively correlated with the anthocyanin content [55]. In this study, the chlorophyll content and ratio in juvenile leaves of *A. tutcheri* were low, the photosynthetic rate was also relatively slow, and the number of carbohydrates synthesized through photosynthesis in juvenile leaves was limited. As the sink system, carbohydrates are preferentially provided for their growth and development, at which time the carbon source in the secondary metabolite pathway is restricted, and the carbon source allocated to the phenylpropanoid pathway for the synthesis of anthocyanin is be reduced, resulting in a decrease in the anthocyanin content.

Soluble proteins are important nutrients and osmoregulatory substances in plants. Most of the soluble proteins in plants are involved in the synthesis of various metabolic enzymes, which are involved in various physiological processes in leaves, such as photosynthesis, signal transduction, pigment metabolism, and carbohydrate conversion [56]. The results of this study show that there was no significant correlation between soluble protein content and each physiological factor during the spring leaf color change of *A. tutcheri*, which is consistent with the findings of Tang [57]. This may be because the various physiological processes during the spring leaf color change are accomplished by the joint participation of multiple enzymes, so the soluble protein content did not differ significantly during any period of the spring leaf color change.

*4.3. Analysis of Physiological Indicators Related to Anthocyanin Degradation during Different Leaf Color Periods*

On the other hand, the higher the POD activity, the lower the anthocyanin content of *A. tutcheri* leaves; correlation and path analysis showed that POD activity had a greater effect on anthocyanin content. As the leaves change from red to green, photosynthesis intensifies, resulting in increased production of reactive oxygen species through reactions such as photo-oxidation and photorespiration [6,58], and excess hydrogen peroxide can enter the vesicles across the membrane [59]. Vacuole peroxidase uses anthocyanin and other phenolics as substrates for coupled oxidation with hydrogen peroxide to reduce the escape of hydrogen peroxide in other organs, and anthocyanin changes color as a result of oxidation [60]. Therefore, POD is considered a possible enzyme related to the degradation of anthocyanin and promotes the degradation of anthocyanin; this was also confirmed in the degradation of anthocyanin in ornamental plants such as *Brunfelsia latifolia* [19], *Malus spectabilis* [61], and *Excoecaria cochinchinensis* [28]. Whereas PPO activity stabilized during the S1 to S4 period and did not significantly correlate with anthocyanin content, this may be because PPO is localized in the cytoplasm rather than co-localized with anthocyanin in the vacuole. Only after cellular regionalization is broken can anthocyanin come into contact with PPO to produce a reaction, and disruption of cellular regionalization implies

senescence or damage from adversity [62]. In addition, Zhao et al. suggested that the degradation of anthocyanin by PPO is related to the structure of anthocyanin and that anthocyanin without an o-diphenol hydroxyl structure was insensitive to PPO [63]. The anthocyanin in *A. tutcheri* may belong to this category. In conclusion, the relationship between the activities of various enzymes in the anthocyanin metabolic pathway and the anthocyanin content needs to be studied.

Because anthocyanins are water-soluble pigments located mainly in the vacuole, their color also depends on the vacuole environment, such as vacuole pH, metal ions, and phenolic compounds [9,24]. In this study, the vacuole pH of *A. tutcheri* gradually increased from the S1 to S4 periods, and both correlation and path analysis showed that the vacuole pH was closely related to anthocyanin content. It is hypothesized that the degradation of anthocyanin in *A. tutcheri* leaves depends on the increase in vacuole pH, which is similar to the findings reported by Zhang et al. on the pH of *Malus spectabilis* leaves [64]. Tannins in phenolic compounds play an important role in anthocyanin coloration [65]. However, there was no significant change in tannin content in the leaves of *A. tutcheri* during the S1 to S4 period (Figure 6). Our results are in contrast to the results reported by Luo et al. on the changes in tannin content during the fading of leaves of *Osmanthus fragrans* and *Excoecaria cochinchinensis*, who documented a significant effect of tannin content in leaves on the stability of anthocyanin [28]. Therefore, the role of tannins in the stability of anthocyanin in *A. tutcheri* leaves needs further investigation.

## 5. Conclusions

In summary, the determination of pigments, microscopic observation, and analysis of multiple physiological factors during the red fading of juvenile leaves of *Acer tutcheri* demonstrated that the S1 period had the highest content and proportion of anthocyanin, and by the S4 period, chlorophyll had rapidly surpassed anthocyanin as the primary pigment in the leaves. This shows that the change in the spring leaf color of *A. tutcheri* is directly related to the quantity and proportion of pigments rather than a change in the type and composition of pigments in the leaves. Osmotic substance measurements showed that there were fewer carbon sources for anthocyanin synthesis in the juvenile leaves of *A. tutcheri*, which resulted in a decrease in anthocyanin content. During spring leaf color fading, it was observed that PAL and CHI activities started to decline during the S2 period, and POD activity increased, revealing their potential role in regulating spring leaf color changes in *A. tutcheri*. It is hypothesized that a combination of a decline in the metabolism of anthocyanin synthesis and an increase in the metabolism of anthocyanin degradation causes the red color to fade in the spring. Our study also revealed that vacuole pH increased during leaf color fading and played a vital role in the decrease in anthocyanin concentration, which may be accomplished by reducing the stability of anthocyanin structures. Our studies show that a combination of physiological factors, such as anthocyanin, chlorophyll, soluble sugars, starch, PAL, CHI, POD, and vacuole pH, works together to cause the change from red to green in the spring leaves of *A. tutcheri*.

**Author Contributions:** Conceptualization, Y.X., B.S., L.C. and N.P.; methodology, Y.X., N.P. and Z.S.; software, Y.X. and Z.H.; validation and formal analysis, Y.X., Z.S. and B.M.; investigation and data curation, Y.X., Q.L., J.L. and M.L.; writing—original draft preparation, Y.X.; writing—review and editing, B.S., N.P., L.C. and Z.S.; visualization, Y.X. and Z.S.; supervision and project administration, B.S. and N.P. All authors have read and agreed to the published version of the manuscript.

**Funding:** This research was funded by the Science and Technology Program of the Forestry Administration of Guangdong Province, China (grant number 2021KJCX017); the National Natural Science Foundation of China (grant number 31600449); and the Program of Guangzhou Municipal Science and Technology Bureau, China (grant number 202102021257).

**Data Availability Statement:** Not applicable.

**Acknowledgments:** The authors thank Tianci Xie for his help with data processing.

**Conflicts of Interest:** The authors declare no conflict of interest.

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
