# Peer review of "Juvenile Leaf Color Changes and Physiological Characteristics of Acer tutcheri (Aceraceae) during the Spring Season"

_forests, doi:10.3390/f14020328_

Round 1
Reviewer 1 Report
The paper is devoted to the investigation of the color performance features and its related physiological properties in the spring leaf growth phase of Acer tutcheri. Authors measured different parameters including content of pigments and cell sap pH, and physiological factors related to anthocyanin metabolism in leaves at 4 stages (soluble sugars content, soluble proteins content, tannin content, phenylalanine ammonialyase activity, chalcone isomerase activity, peroxidase and polyphenol oxidase activities). Authors tried to determine the major physiological factors in leaf color changes.
There are several mistakes and not clear moments in the MS.
Lines 97 and 98. The sentence “Gradual increase in the pH of petal vacuole during the degradation of anthocyanin in Nelumbo nucifera [27]” has no connection with entire text and has no verb (predicate).
The interval in sampling (15 days) should be explained. Why not every day or every week?
9 biological replicates is a good option. It is not clear why authors applied pooling of the samples and used 3 biological replicates instead of 9. Should be explained. Why leaves from 3 different trees were mixed?
Line 132. Mistake. Correct ‘soution’ to ‘solution.
Line 138. ‘extracted from the light’? What does it mean?
Lines 142-143. Autors write ‘Calculate the concentration and content of carotenoid (Car), total chlorophyll 142 (Chl), chlorophyll a (Chl a), and chlorophyll b (Chl b)’. Calculation should be described in details, formula or link to the publication should be provided.
Lines 150-151. The same for “Calculate the concentration and content of anthocyanin (Ant)”. Calculation should be described in details, formula or link to the publication should be provided.
Methods of the ‘soluble sugars content, soluble proteins content, tannin content, phenylalanine ammonia-lyase activities, chalcone isomerase activities, peroxidase activities and polyphenol oxidase activities ‘ analysis should be described or catalog numbers of the Kits should be mentioned.
Line 160: “Making of slices and staining”. Slices of what? Should be specified.
The description of the methods should be in past tense.
What is ‘neutral gum’ (line 176)?
The sentence “Studies on the specific connection between the distribution and content of anthocyanin in juvenile leaves and the properties of photosynthetic leaves can be performed later” looks strange. This was not the aim of the study. Of course this is interesting and may be studied additionally but it is not necessarily to write about it.
What is ‘observation of slice’? Should be change to the scientific lexicon. Microscope analysis or microscopic observation.
The main conclusion is ‘Acer tutcheri's leaves changed from red to green in the spring as a direct result of a decrease in anthocyanin levels and an increase in chlorophyll accumulation’ is so strange. Like grass is green because of chlorophyll. The main conclusion should be rewritten. Tha main accent in the conclusion should be focused on the detected changes in studied parameters during the process of leaves color change.
Athors write following “This study shed a light on the formation of different colored leaves of Acer tutcheri, which also provides a reference for reasonable cultivation measures to improve the decorative characteristics and regulate the timing of colorful leaves”.
Based on the obtained data, how can we regulate the timing of leaves colour changes?
Author Response
Thanks for your professional review work on our article. The point-to-point responses to your comments are attached.
Please see the attachment.

Reviewer 2 Report
Juvenile leaf color changes and physiological characteristics of Acer tutcheri (Aceraceae) during the spring season
forests-2152043
Dear Editor,
Thank you very much for choosing me as a reviewer. I read the MS mentioned some comments and suggestions. I believe that this work is good work and authors have done their best to produce a reliable and practical outcome. However, there are some grammatical errors that need to be addressed. I quit the check.
Keywords: Please add a word related to the Fading OR Discoloration.
L. 50: There is no “apostrophe s” in the academic writing.
2. Materials and Methods
2.1 Experimental Design
All abbreviations at first should be presented in full form and then for next coming, present in abbreviation form.
L. 117-118: Were these trees planted in? Were they on the roadside, in Parks, Forest plantation OR somewhere else?
L. 120: What do you mean by “typical sampling method”? This needs to describe more. How did you sample?
L. 119-120: Please define that the leaf sampling was done from which part of the tree crown. Upper/down/middle/ inside the branches or…?
L. 131: FAA fixative solution (45% anhydrous ethanol, 6% acetic acid, and 5% formaldehyde+46% double-distilled water). Plz, define the reference that was used.
L. 132: check the word solution.
L. 134: Chlorophyll (Chl) and carotenoid (Car) were determined according to Lichtenthaler & Wellburn [29,30]. was used as a reference.
L. 135: “The leaves were cleaned with distilled water, dried, cut, and mixed.” Plz, describe the way you dried, cut and mixed. How did you do? The methods and techniques should be explained.
L. 142: the sentence is grammatically incorrect. “Calculate the concentration…”.
L. 144-151 should be improved based on the comments that came from the above paragraph.
Section 2.2. Plz indicate the replications for each physiological trait.
L. 160: Preparation of … instead of “2.3 Making of slices and staining”
L. 228: What do you mean by “accounting for 93.8%, when (Chl + Car) accounted for only 6.2%?
L. 270. Section 3-3: Where did you define POD and PPO in the Materials and Methods Section? I couldn’t find it. The authors should define all variables and abbreviations before the Results Section.
L. 271-272: you are not allowed to use references in the Results Section. Correct them, please.
Discussion
L. 343: Plz refer these results to the related Table/Figure.
L. 346: better to omit “and other tree species [41]. I am sure in the ref. 41, the other tree species were named.
L. 375: Plz refer your result to the related Fig/Table and so on in the Results Section.
l. 415: you are not allowed to use “etc” in academic writing to state “and so on” and “the rest”. Plz omit it.
L. 417: Plz refer your result to the related Fig/Table and so on from the Results Section.
Author Response
We sincerely thank you for your constructive comments on our manuscript. The point-to-point responses to your comments are attached.
Please see the attachment.

Reviewer 3 Report
Dear Authors,
In this study, the authors aimed to shed a light on the formation of different colored leaves of Acer tutcheri, which also provides a reference for reasonable cultivation measures to improve the decorative characteristics and regulate the timing of colorful leaves. Authors show that the combination of physiological factors, such as anthocyanin, chlorophyll, soluble sugars, starch, PAL, CHI, POD, and vacuole pH work together to cause the change from red to green in the spring leaves of Acer tutcheri.
A few minor revisions are marked in the attached file.

Author Response
We are very grateful to the reviewer for reviewing the paper so carefully. The point-to-point responses to your comments are attached.
Please see the attachment.

Reviewer 4 Report
The following research article focuses on the physiological determinism of juvenile leaf color of the ornamental plant; Acer tutcheri (Aceraceae) during the spring season. As mentioned by the authors, leaf color changes have only been investigated for autumn leaves. No previous work was conducted on spring leaves for this plant species. However, similar studies were conducted on other plant species belonging to Acer genus, such Acer rubrum (Huan et al., 2022). Thus, it would be more interesting to study the molecular aspect of color change in this species, since the physiological and biochemical analyses have already been described in the literature.
I would recommend authors focus more on the molecular aspect of leaf color changes occurring during the spring season since it is evident that at the physiological level, this process is governed by the changes in the following physiological indicators: chlorophyll, carotenoids, and anthocyanin.
Minor recommendations:
Figure 1 and Figure 4 contents are not clear. Please increase the quality and the size of the figures (more precisely, histological observations pictures)
There are some typos in the text. I would recommend a thorough reading of the whole manuscript.
Author Response
We have carefully considered the suggestion of the reviewer, and we made some changes to our manuscripts. The point-to-point responses to your comments are attached.
Please see the attachment.

Round 2
Reviewer 2 Report
Dear Authors,
I appreciate your effort and tact in doing this good work.
Author Response
Dear reviewer,
Thank you for reviewing our manuscript again. According to your suggestions, we have made the following revisions to English through Dr. Zhaowan Shi, from the Department of Fruit Tree Sciences, Plant Sciences, Agricultural Research Organization, volcani center, Israel.
We sincerely appreciate all your constructive and positive comments.
Kind regards,
Dr. Lei Chen

Reviewer 4 Report
The present work focuses on the physiological determinism of leaf color in spring-season plants of the Acer genus. The authors addressed all the previous remarks and made significant changes to the manuscript which improved in a substantial way the quality of the conducted research. I would recommend its publication in the journal at the current form.
Author Response

(The authors gave the same response as above.)
